# Time-Dependent Strength Behavior, Expansion, Microstructural Properties, and Environmental Impact of Basic Oxygen Furnace Slag-Treated Marine-Dredged Clay in South Korea

**Gyeong-o Kang [1], Jung-goo Kang [2], Jin-young Kim [3] and Young-sang Kim [4],***

[1] Department of Civil Engineering, Gwangju University, 277 Hyodeck-ro, Nam-gu, Gwangju 61743, Korea; erosion1004@gmail.com

[2] Underground Space Safety Research Center, Korea Institute of Civil Engineering and Building Technology, Goyangdae-ro 283, Ilsanseo-Gu, Goyang-si 10223, Korea; kangjunggoo@kict.re.kr

[3] Construction Automation Research Center, Korea Institute of Civil Engineering and Building Technology, Goyangdae-ro 283, Ilsanseo-Gu, Goyang-si 10223, Korea; goldcamp@kict.re.kr

[4] Department of Civil Engineering, Chonnam National University, 77 Yongbong-ro, Buk-gu, Gwangju 61186, Korea

* Correspondence: geoyskim@jnu.ac.kr

**Abstract:** The aim of this study was to investigate the mechanical characteristics, microstructural properties, and environmental impact of basic oxygen furnace (BOF) slag-treated clay in South Korea. Mechanical characteristics were determined via the expansion, vane shear, and unconfined compression tests according to various curing times. Scanning electron microscopy was conducted to analyze microstructural properties. Furthermore, environmental impacts were evaluated by the leaching test and pH measurements. According to the results, at the early curing stage (within 15 h), the free lime (F-CaO) content of the BOF slag is a significant factor for developing the strength of the adopted sample. However, the particle size of the BOF slag influences the increase in the strength at subsequent curing times. It was inferred that the strength behavior of the sample exhibits three phases depending on various incremental strength ratios. The expansion magnitude of the adopted samples is influenced by the F-CaO content and also the particle size of the BOF slag. Regarding the microstructural properties, the presence of reticulation structures in the amorphous gels with intergrowths of rod-like ettringite formation was verified inside the sample. Finally, the pH values and heavy metal leachates of the samples were determined within the compatible ranges of the threshold effect levels in the marine sediments of the marine environment standard of the Republic of Korea.

**Keywords:** basic oxygen furnace slag; strength behavior; expansion; microstructural properties; environmental impact; marine-dredged clay

## 1. Introduction

Steelmaking slags originate from steelmaking refining processes, in which impurities (e.g., carbon, phosphorus, sulfur, silicon) are eliminated from the molten steel [1,2]. In addition, based on the kind of furnace adopted (converter or electric arc furnaces), steelmaking slags can be generally categorized into two types: basic oxygen furnace (BOF) and electric arc furnace (EAF) slags [1,2].

In South Korea, approximately more than 10 million tons of steelmaking slags are produced annually, and owing to their high friction angle, abrasion resistance, and high stiffness, they are mostly recycled as civil and road construction aggregates [3,4]. However, this application of slags is limited to slags with low or completely eliminated free lime (F-CaO) and free periclase (F-MgO) content via the aging treatment process. This can be attributed to the hydration of F-CaO and F-MaO with water, which results in volumetric instability, i.e., an expansion that approximately doubles the volume [5,6].

It is well known that the chemical composition and mineralogy properties of BOF slag are similar to those of Portland cement. The mineralogical composition of BOF slag is rhombohedral to orthorhombic (R-O) phase (solid solution of CaO, FeO, MgO, and MnO), dicalcium silicate ($C_2S$), tricalcium silicate ($C_3S$), tetra-calcium aluminoferrite ($C_4AF$), dicalcium ferrite ($C_2F$), F-CaO, olivine, and merwinite [7–9]. Furthermore, some researchers have stated that BOF slag is considered a weak Portland cement with a low hydraulic activity primarily owing to its low tricalcium silicate content ($C_3S$) and high iron oxide content [10–12]. Hence, they suggested that BOF slags can function as a binder to stabilize soils to achieve low strength without activators. It has been reported that under aqueous conditions, the hydraulic activity of slag increases with an increase in the alkalinity of certain slag types, such as $CaO/SiO_2$ and $CaO/(SiO_2 + P_2O_5)$ [13,14]. According to previous studies [7–14], to obtain soil stabilization, ground BOF slag was adopted as a binder together with different activators, and these studies confirmed significant improvements in strength and durability owing to the improvements in the hydraulic reactivity. Based on the literature review, the chemical and mineralogical properties as well as the alkalinity of slags are possibly the major factors for determining hydraulic reactivity.

Japan has published a guidebook for the oceanographic application of converter BOF slags. The guidebook recommends that after augmenting their grain sizes and compositions, BOF slags can be used as stabilizers for marine-dredged clay (MDC), and they are called calcium-oxide-based reformers. Adjusted BOF slags mixed with marine clay, dredged from maintaining the navigation channel and securing the harbor and its mixture, have been used as civil construction materials that require relatively low strength, such as the filling and reclamation materials in quay walls, submerged embankments, and breakwater [2,15,16]. This application technique in both BOF slag and MDC is significantly beneficial from economic and environmental perspectives. The strength mechanisms of BOF-treated clay were significantly influenced by two reactions, hydration and pozzolanic reactions, which generate the gels of the compound in a clay fabric [15,16]. Toda et al. [17] reported that C-S-H gels belong to the secondary phases (pozzolanic reaction) that are responsible for the enhancing the strength of BOF slag-treated clay, and the amounts of amorphous silica in the dredged clay, as well as the amount of portlandite ($Ca(OH)_2$) in the BOF slag, are the major key factors influencing the formation of C-S-H gels of the BOF-treated dredged clay.

Kang et al. [2] stated that in Japan, the strength development of the BOF slag-treated clay comprising the BOF slag and dredged clay is altered with specific curing times, ranging from significantly short-term curing (0.5 h) to long-term curing (90 days). In addition, they proposed time-dependent equations for estimating clay strength based on the volumetric solid content, initial water content, and strength increment coefficients. According to Cikmit et al. [18], the strength of the BOF slag-treated dredged clay is affected by the maximum particle size of the BOF slag under the same free lime content. However, most of the previous studies related to the BOF slag have been focused on the application of BOF slag as an aggregate material in civil works and road construction, or as binder additives [3,4,19–27]. Furthermore, a few studies exist on the engineering and physicochemical properties of BOF-treated clay with various types of BOF slags and MDCs. Consequently, the effect of the components of BOF slag and dredged soils on strength development remains unclear.

In this study, the time-dependent strength behavior, expansion, microstructural properties, and environmental impact of a mixture of BOF slag and MDC generated in South Korea were investigated. Samples were prepared with two types of BOF slag and one type of MDC with different BOF slag and water content. To evaluate the engineering and microstructural properties and environmental impacts, laboratory vane shear (LVS) and unconfined compression (UC) tests and an expansion test, as well as scanning electron microscopy (SEM), leaching tests, and pH measurement, were conducted, and the results obtained were analyzed.

## 2. Materials and Methods

### 2.1. Materials

The MDC samples used in this research were obtained from Busan Port, South Korea, and adopted as the base clay material in all experiments. The physical properties of the adopted MDC are comprehensively presented in Table 1. In addition, the MDC samples mostly consisted of fine-grained soils (less than 75 μm) and were divided into elastic clay or organic clay with high plasticity (CH or OH) according to the Unified Soil Classification System (USCS).

**Table 1.** Physical properties of dredged marine clay at Busan Port.

| Property | Value |
|---|---|
| Liquid limit, $w_{LL}$ (%) | 67.29 |
| Plastic limit, $w_{PL}$ (%) | 36.44 |
| Plasticity index, $w_{PI}$ (%) | 30.84 |
| Specific gravity, $G_s$ (g/cm$^3$) | 2.64 |
| Coarse-grained soil (%) | 5.27 |
| Fine-grained soil (%) | 94.73 |
| Unified Soil Classification System (USCS) | CH/OH |
| Initial water content (%) | 61.61 |

The BOF slags adopted in this study were supplied by two ironworks in South Korea and are hereafter referred to as BOF A and BOF B. The physical properties of the adopted BOF slags are summarized in Table 2. The F-CaO contents of BOF A and BOF B slags that substantially influence strength development were determined via an ethylene glycol method, and the BOF A and BOF B slags exhibited 4.48% and 1.68% F-CaO contents, respectively. The F-CaO content of the BOF A slag was higher than that of the BOF B slag. In addition, although both BOF slags were applied at less than 5 mm, in the median grain size ($D_{50}$) case, the BOF B slag ($D_{50}$ = 0.77 mm) was smaller than that of BOF A ($D_{50}$ = 2 mm). The particle size of the BOF slag could possibly be a significant factor related to the strength increase owing to the specific surface area that triggers high reactivity. Figure 1 presents the particle-size distribution curves for the adopted MDC and BOF slags.

The elemental compositions of the BOF slags measured via X-ray fluorescence (XRF) MiniPal 2/PANanalytical (Netherlands) are listed in Table 3 and compared to those of other BOF slags obtained from previous studies [7,11]. The elemental compositions of the adopted BOF slags mainly consist of silicon (SiO$_2$), calcium (CaO), and iron (Fe$_2$O$_3$) oxides, accounting for approximately 80%, and are similar to those of previous studies.

**Table 2.** Physical properties of basic oxygen furnace (BOF) slag.

| Property | Value | |
|---|---|---|
| | **BOF A** | **BOF B** |
| Saturated surface dry density (g/cm$^3$) | 3.66 | 3.45 |
| Absolute dry density (g/cm$^3$) | 3.52 | 3.31 |
| Water absorption rate (%) | 3.89 | 4.26 |
| Initial water content (%) | 3.49 | 2.22 |
| Median grain size, $D_{50}$ (mm) | 2.00 | 0.77 |
| Coarse-grained soil (%) | 98.53 | 93.97 |
| Fine-grained soil (%) | 1.47 | 6.03 |
| Free calcium, F-CaO (%) | 4.48 | 1.68 |

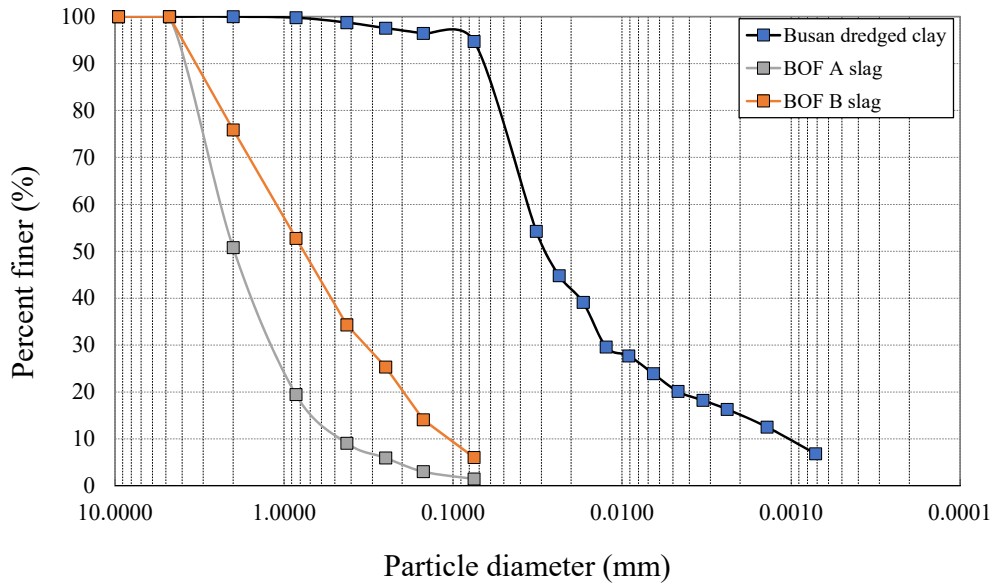

**Figure 1.** Particle-size distribution curves for marine-dredged clay (MDC) and BOF slags.

**Table 3.** Elemental compositions of BOF slags.

| Chemical | BOF A | BOF B | BOF (%) [a] | BOF (%) [b] |
|---|---|---|---|---|
| $SiO_2$ | 16.4 | 24.3 | 8–20 | 10.8–13.1 |
| CaO | 35.2 | 36.6 | 30–55 | 40.1–45.0 |
| $Fe_2O_3$ | 29.8 | 16.4 | 10–35 | 28.3–32.0 |
| MgO | 6.4 | 6.8 | 5–15 | 4.5–7.5 |
| $Al_2O_3$ | 3.4 | 9.7 | 1–6 | 1.7–2.1 |
| MnO | 3.1 | 1.7 | 2–8 | 2.0–3.7 |
| $TiO_2$ | 0.8 | 0.6 | 0.4–2 | 0.5–0.9 |
| $P_2O_5$ | 2.4 | 1.5 | 0.2–2 | 1.4–2.4 |
| $SO_3$ | 0.3 | 1.0 | 0.05–0.15 | 0.4–1.2 |
| $Na_2O$ | 0.7 | 0.4 | N/A [c] | N/A [c] |
| $K_2O$ | 0.3 | 0.4 | N/A [c] | N/A [c] |
| Others | 1.2 | 0.6 | N/A [c] | N/A [c] |

[a] Shi [11], [b] Belhadj [7], [c] not available.

Sample Preparation

To obtain a homogeneous mixture, coarse particles and impurities in the MDC samples were thoroughly eliminated by a 2 mm sieve. The setting initial water content ($w_0$) of the samples was calculated as a ratio of the liquid limit ($w_{LL}$) of the DMC, which is expressed as $w_0/w_{LL}$ = 1.2, 1.5, and 2.0. In addition, the additional water required for setting the initial water content was prepared as artificial seawater with a 3.5% salinity. However, the BOF slags were used after they were fully air-dried in a room for one day at a temperature of 20 ± 3 °C to achieve accurate water content setting in all experiments of this study by removing the initial water content of slag. In addition, the BOF slag content, $BOF_{vol.}$, was determined using a volumetric ratio equation expressed as:

$$BOF_{vol.} = \frac{V_{BOF}}{V_{BOF} + V_{soil} + V_{water}} \times 100 \ (\%) \tag{1}$$

where, $V_{BOF}$, $V_{soil}$, and $V_{water}$ represent the solid volume of the BOF slag, solid volume of the MDC, and volume of water, respectively.

For sampling, pre-prepared BOF slag and additional water were mixed with MDC slurry and thoroughly stirred for 10 min using a hand mixer. The mix proportions and curing time conditions for the samples adopted in the experiments are presented in Table 4.

Here, after sealing the samples with polythene wrap, the samples were immediately cured in a curing chamber at a temperature and humidity of $20 \pm 3\,°C$ and $95 \pm 2\%$, respectively.

**Table 4.** Mix proportions and curing conditions of performed tests.

| Type of Test | Setting Initial Water Content ($w_0/w_{LL}$) | BOF Content, $BOF_{Vol.}$ (%) | Curing Time |
|---|---|---|---|
| LVS and UC * | 1.2, 1.5, 2.0 | 20, 30, 40 | 0.5, 2, 5, 7, 10, 15 (h) 1, 2, 3, 7, 28, 90, 150 (days) |
| Expansion | 1.2 | 20, 30, 40, 60, 80, 100 | 0 to 90 days |
| SEM | 1.2 | 40 | 28 and 90 days |
| pH | 1.2, 1.5, 2.0 | 20, 30, 40, 100 | 7, 14, 28, and 90 days |
| Leaching | 1.2 | 20, 40 | 90 days |

* LVS: laboratory vane shear; UC: unconfined compression.

### 2.2. Methods

Comprehensive experimental programs measuring characteristics such as strength behavior, expansion, and microstructural properties, as well as environmental impact, were performed in various test conditions. For strength measurement, LVS and UC tests were conducted based on the curing time conditions: short-term to long-term curing periods. In addition, an expansion test was conducted for the samples with or without clay to evaluate the impact of adding clay for expansion. To observe the cementitious compounds and interparticle contact in the clay fabric, SEM analysis was investigated with different BOF slags and water contents. Regarding the environmental impacts on the treatment of BOF slag in MDC, a leaching test of heavy metals and pH test of corrosivity were conducted on the samples with or without BOF slag treatment, as well as on the influence of the increase in BOF slag content.

#### 2.2.1. LVS and UC Tests

As presented in Table 4, the curing condition of the set of strength tests from very-short-term to long-term curing times ranged from 0.5 h to 150 days. The LVS test was employed for the sample at a very-short-term curing time, i.e., up to 15 h or 1 day, based on the sample condition. The preparation and instructions for the adopted LVS test were based on [28]. The sample that allowed sufficient clearance of the blade diameters above was prepared, and the sample diameter was 10 cm. In this test, the height and diameter of the vane employed were 4 and 2 cm, respectively, such that the height of the vane was twice the diameter (H/D = 2:1). The torque spring device was rotated at a constant rate of $60°/min$ until the spring deflection stopped increasing.

According to [29], the UC test was performed for the remaining samples with curing times from 15 h to 150 days. Cylindrical specimens, 100 mm in height and 50 mm in diameter (H/D = 2), were prepared, and the load was applied with an axial strain at a rate of 2%/min until the load values decreased or 15% strain was reached. The unconfined compressive strength ($q_u$) was determined as the compressive stress at the point of failure.

The $s_u$ was conventionally calculated to be 1/2 of the $q_u$. Therefore, in this study, to equate the values of strength magnitude obtained from the LVS and UC tests, $q_u$ was recalculated to be two times $s_u$ ($2s_u = q_u$).

#### 2.2.2. Expansion Test

The expansion test was conducted according to [30], based on the potential expansion of aggregates from the hydration reaction. The cylindrical sample, with height and diameter of 175 and 150 mm, respectively, was prepared according to the general procedures of the California bearing ratio (CBR) after compaction was performed with maximum density. To accelerate the hydration reaction, a surcharge of 4.54 kg was added to the specimen and submerged in a water bath at a temperature of $70 \pm 3\,°C$. The expansion percentage was

calculated by dividing the difference between the daily dial gauge and base readings by the initial height of the specimen.

### 2.2.3. SEM Analysis

For the investigation and comparison of microstructure properties, such as the cementitious and chemical compositions of the samples comprising different types and contents of BOF slag, a Hitachi S-3000N scanning electron microscope was employed to perform microstructure analysis. To obtain optimal images and results for the analyses, the sample was prepared by freeze-drying and coated with gold.

### 2.2.4. pH and Heavy Metal Leaching Tests

To assess the safety and environmental impact of BOF slag-treated clay with different types and contents of BOF slag, as well as curing time, pH measurements and a heavy metal leaching test were conducted. To fabricate the leachate of a sample, the sample was immersed in demineralized water at a water-to-solid volume ratio (W/S) of 8 at a temperature of 21 °C (W/S = 8). The pH was then measured in leachates for various samples of raw BOF slags and BOF slag-treated clays.

Regarding the heavy metal leaching test, the sample was completely submerged in demineralized water at W/S = 8 for a given number of days, and the collected water was filtered using a vacuum device to remove the coarse particles in the sample. After that, the filtered water was used as a leachate to determine the heavy metal concentration. Ultimately, the leachates were examined for various samples of raw BOF slags and BOF slag-treated clays via inductively coupled plasma atomic emission spectrometry (ICPMS).

## 3. Results and Discussion

### 3.1. Time-Dependent Strength Behavior

Identifying the strength development process required at various curing times immediately after mixing is beneficial for the design of civil works, such as submerged embankments and reclamation [31,32]. Consequently, by considering time-dependent behavior from immediately after mixing to long-term curing times, several models have been proposed for estimating the strength of MDC treated by Portland cement and BOF slag [2,31,32]. Figure 2 presents the strength behavior with curing times starting from immediately after mixing the BOF slag with various samples treated with different BOF slag and water contents at a log–log scale. All samples exhibited an increase in time-dependent strength. In addition, the time-dependent strength of all samples increased linearly over the curing times regardless of the BOF slag setting and initial water contents. As presented in Figure 2, the strength growth of the BOF slag-treated clay passed through three phases relative to curing time, with different incremental strength ratios: phase I (curing time: 0.5 to 10 h), phase II (curing time: 10 h to 3 days), and phase III (curing time: 3 to 150 days).

Notably, although the three phases in strength development categorized by the curing time observed in this study are similar to those of previous studies, the starting point of the curing time corresponding to each phase is different, as illustrated in Figure 3. Kang et al. [2] reported that the strength development of the BOF slag-treated clay in Japan was classified into three zones: inactive (0.5 to 5 h), active (5 h to 3 days), and moderate zones (3 to 90 days). Therefore, the starting points of strength development at the inactive and active zones corresponding to phases I and II in this study are 0.5 to 5 h and 5 h to 3 days, respectively. In addition, the strength behavior at the curing time immediately after mixing the Portland-cement-treated clay can be divided into two stages: before and after 3 days of curing [31]. Hence, the strength behavior of the stabilized clay with curing time can be altered by the types of stabilizers and marine-dredged clays adopted. The strength of all BOF slag-treated clay samples increased with an increase in the BOF slag content under the same initial water content condition and decreased with an increasing initial water content under the same BOF slag content condition. Furthermore, it was determined that the magnitudes of strength value are developed differently depending on

the type of BOF slag and range of curing time. The strength values in BOF A slag-treated samples were, at a maximum, approximately 3.7 times higher than those of the BOF B slag-treated samples under the same condition at early curing times, ranging from 0.5 to 15 h, except for three data. In contrast, at a later curing range, the BOF A slag-treated samples showed strength values approximately 0.25 times lower than those of the BOF B slag-treated samples, except for some data.

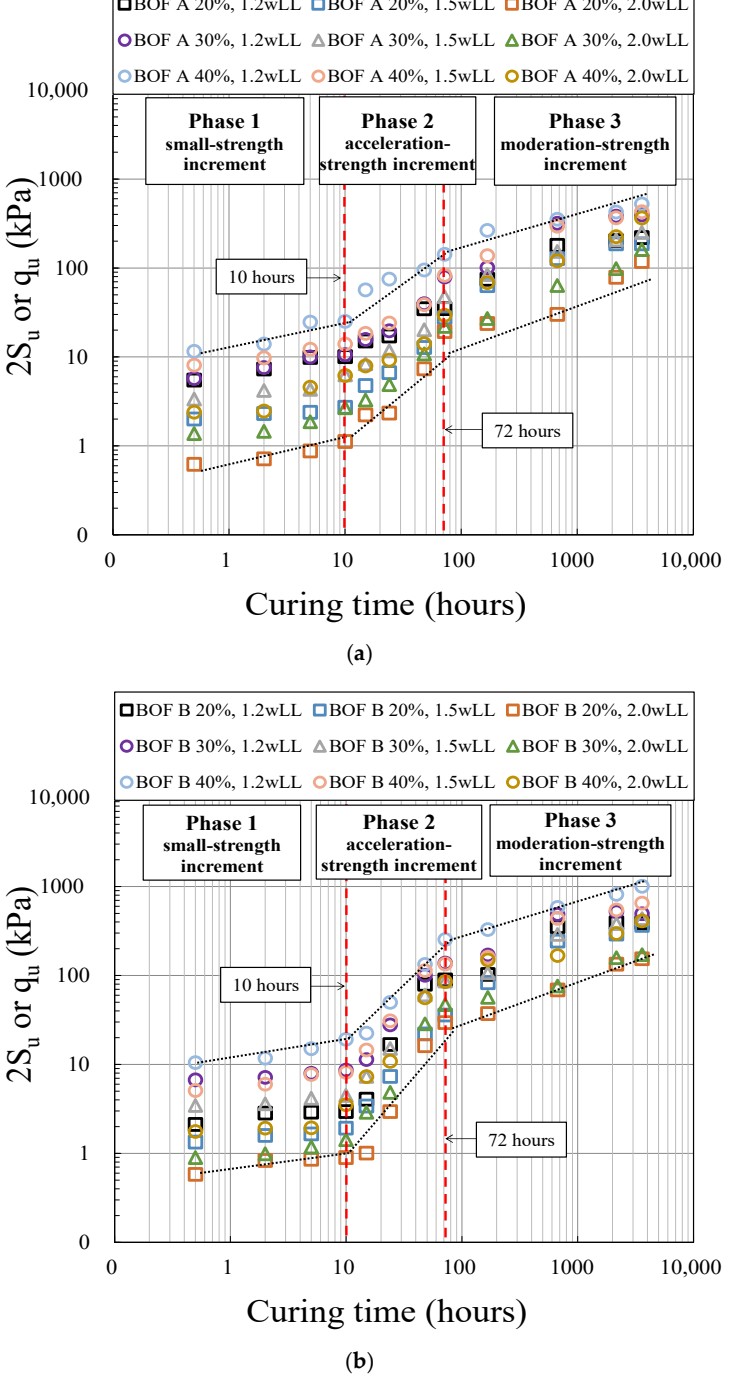

**Figure 2.** Time-dependent strength behavior for (**a**) BOF A and (**b**) BOF B slags.

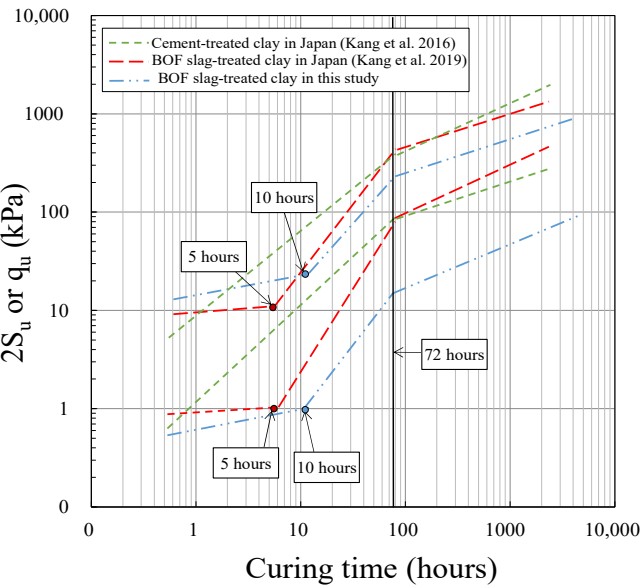

**Figure 3.** Comparison of time-dependent strength behavior of different types of BOF slags and stabilizers.

In addition, it was observed that the incremental strength ratio of each phase is different. Initially, the strength development was slowest at phase I (small-strength increment), then it abruptly increased at phase II (acceleration-strength increment), and finally slowed down at phase III (moderation-strength increment). The incremental strength ratios for all adopted samples were calculated by $\Delta(\log q_u)/\Delta(\log t)$, and were expressed as $\beta_1$, $\beta_2$, and $\beta_3$ for each phase, respectively. Figure 4 presents the $\beta$ value according to the type of BOF slag adopted. The $\beta$ exhibited a reduction with a decrease in the BOF slag content, as well as an increase in the initial water content for all used samples. In the phase I case, $\beta_1$ of the BOF A slag ranged from 0.05 to 1.51. Moreover, the $\beta_1$ of the BOF B slag ranged from 0.03 to 0.96. At phase II, the $\beta_2$ values of the BOF A and BOF B slags were from 0.28 to 1.64 and 0.48 to 3.84, respectively. For phase III, the $\beta_3$ values of the BOF A and BOF B slags were from 0.48 to 3.84 and 0.28 to 1.64, respectively. Comparing the $\beta$ values of the BOF A and BOF B slags, it can be inferred that a significant variation exists in the $\beta_2$ value of phase II (acceleration-strength increment) depending on the type of the BOF slag. In addition, it was observed that the $\beta$ values are altered based on the phases mentioned in this study. The $\beta_1$ value of the BOF A slag at phase I was larger than the $\beta_2$ and $\beta_3$ values of the BOF B slags at phases II and III. In other words, at phase I, the strength development of the BOF A slag is faster than that of the BOF B slag, whereas the strength development of the BOF B slag is faster than that of the BOF A slag in phases II and III.

The possible reasons for these results might be the effects of median grain size ($D_{50}$), fine particle content (*FC*), and F-CaO content. As earlier mentioned, at phase I, the strength development of the BOF A slag-treated clay was superior to that of the BOF-B-treated clay. Hence, the $D_{50}$ and F-CaO content could possibly be responsible for the strength development of the BOF slag-treated clay at the early curing stage. In detail, the strength development immediately after mixing—0.5 h curing—was governed by a larger particle size owing to the increase in the effective particle contact area. In addition, the F-CaO content is an important property of the incremental strength ratio due to the hydration reaction at early curing, as defined by $CaO + H_2O = Ca(OH)_2$. Meanwhile, for phases II and III, the *FC* and $D_{50}$ possibly play dominant roles in improving the strength development of the BOF-treated clay. The strength development of the BOF slag-treated clay at long-term curing is attributed to the formation of cementitious hydrates, such as the C-S-H and C-A-H gels produced by the reaction between $Ca(OH)_2$ in the BOF slag, and silica (Si) and alumina (Al) presented in MDC. The chemical formula for this reaction is given as $Ca(OH)_2 + [SiO_2, Al_2O_3] \rightarrow$ C-S-H, C-A-H gel. A possible reason for this result is that the smaller

particle size evidently has a specific surface area with high reactivity that produces more cementitious hydrates.

Note that in this study, the strength development for the effects of $C_2S$, $C_3S$, and $C_4AF$ contained in the BOF slag was ignored owing to their negligible amounts [16].

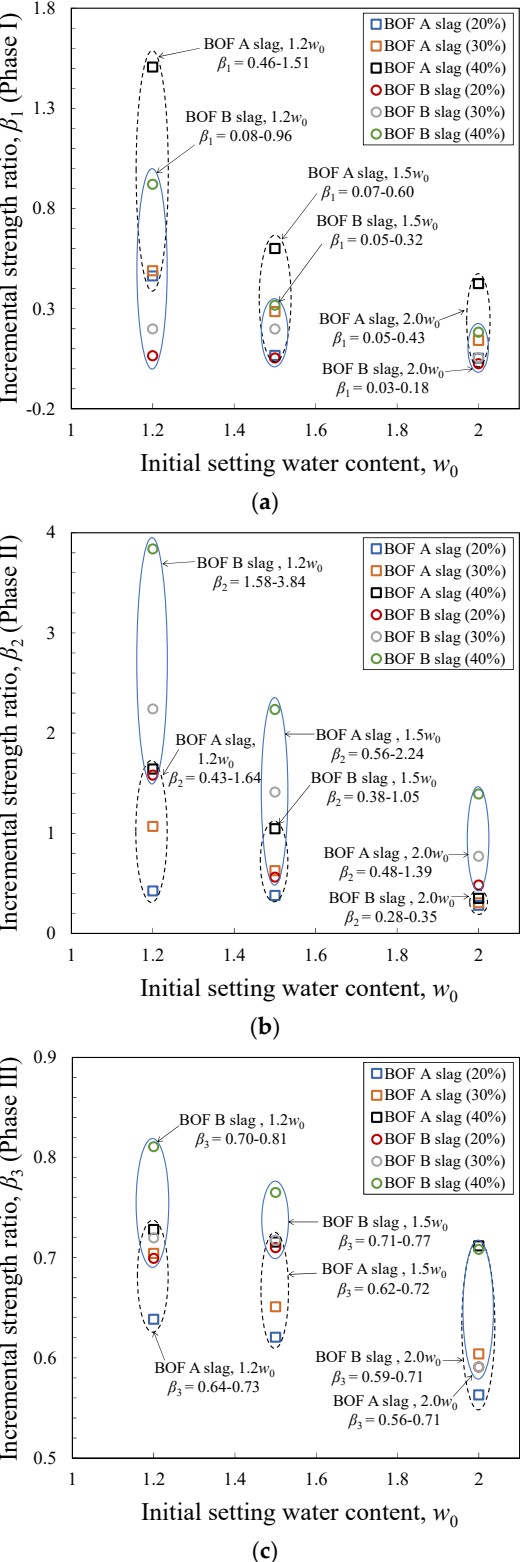

**Figure 4.** Incremental strength ratios based on phases: (**a**) phase I (small−strength increment), (**b**) phase II (acceleration−strength increment), and (**c**) phase III (moderation−strength increment).

### 3.2. Volumetric Expansion Properties

The F-CaO and F-MgO contents primarily responsible for volume instability are one of the most critical challenges facing the recycling of BOF slag aggregates in construction. Therefore, it is crucial to determine the amounts of F-CaO and F-MgO required. In addition, the expansion properties of the F-CaO and F-MgO contents in BOF slag can be divided into two categories: short-period and long-period expansions [33]. The F-CaO content is a dominant factor of the short-period expansion (a few days). In addition, the long-period expansion is controlled by F-MgO. In particular, the hydration reaction of F-MgO can occur over a long period and continues for a few years [33]. Therefore, based on previous studies, it was inferred that the hydration reaction of F-CaO is faster than that of F-MgO with large variations in volume. Generally, the hydration reaction generated when F-CaO and F-MgO are exposed to water can be expressed in Equation (2):

$$CaO \text{ and } MgO + H_2O \rightarrow Ca(OH)_2 \text{ and } Mg(OH)_2 \tag{2}$$

The hydrations of F-CaO and F-MgO trigger volume increases of approximately 91% and 120%, respectively [33]. However, the hydration of F-MgO in the BOF slag can only occur in low basicity conditions and is volumetrically unstable [33].

Note that in this study, the expansion generated by F-CaO in the BOF slag is considered as a major contributing factor [6,13].

Figure 5 shows the expansion percentages based on the BOF slag/clay content ratios with different types of BOF slags adopted in this study. Generally, the expansion percentage of employed samples increased with an increase in the BOF slag content and time. The maximum expansion ratios of raw BOF A and BOF B slags were 3.97% and 8.72%, respectively. In addition, the end time of expansion for the raw BOF A slag was approximately 20 days, which is faster than that of the raw BOF B slag of approximately 80 days. It was found that although the F-CaO content of the raw BOF A slag was 4.48%, which was higher than that of the raw BOF B slag (1.68%), and the expansion ratio of the raw BOF B slag was approximately 2.2 times larger than that of the raw BOF A slag. Comparing both BOF slags adopted, the significant differences between the BOF A and BOF B slags were the mean particle size and particle size distribution. According to previous studies [34–36], the expansion by the alkali–silica reaction can be changed with the aggregate size and particle size distribution. Shu and Kuo [36] carried out expansion tests for various concrete specimens, including various sizes (0.15−4.75 mm) and quantities of BOF slags (10−25%), and reported a specific BOF slag size that exhibits maximum volumetric expansion under the same F-CaO content. Based on the results reported in the literature, a possible reason for the expansion ratio results in this experiment could be the effects of the particle size and particle size distribution of the BOF slag.

Figure 6 illustrates the expansion behavior of the BOF-treated clay according to its clay content. From this figure, it is revealed that the expansion behavior of the BOF-treated clay is altered with changes in the BOF slag and clay contents. The expansion of the raw BOF slag was significantly high compared with the BOF slag-treated clay; however, the expansion abruptly decreased to less than 1% up to a certain clay percentage after adding clay. In detail, the expansion of the BOF slag-treated clay decreased with an increase in the BOF slag until the addition of clay increased by 40%, after which no expansion occurred until the percentage of clay was 70%. Interestingly, expansion begins to increase from high clay content, ranging from 70% to 100%. This means that the expansion in the BOF-treated clay can be triggered by both the BOF slag and also the clay, beyond a specific content. Therefore, the dominant factors, i.e., BOF slag or clay, which contribute to the expansion of the BOF-treated clay, can be altered according to the BOF slag and clay contents. Based on the results obtained from this study, the expansion behavior of the BOF-treated clay can be classified into three zones with different expansion percentages: expansion zones 1 (effect of slag), 2 (no effect), and 3 (effect of clay).

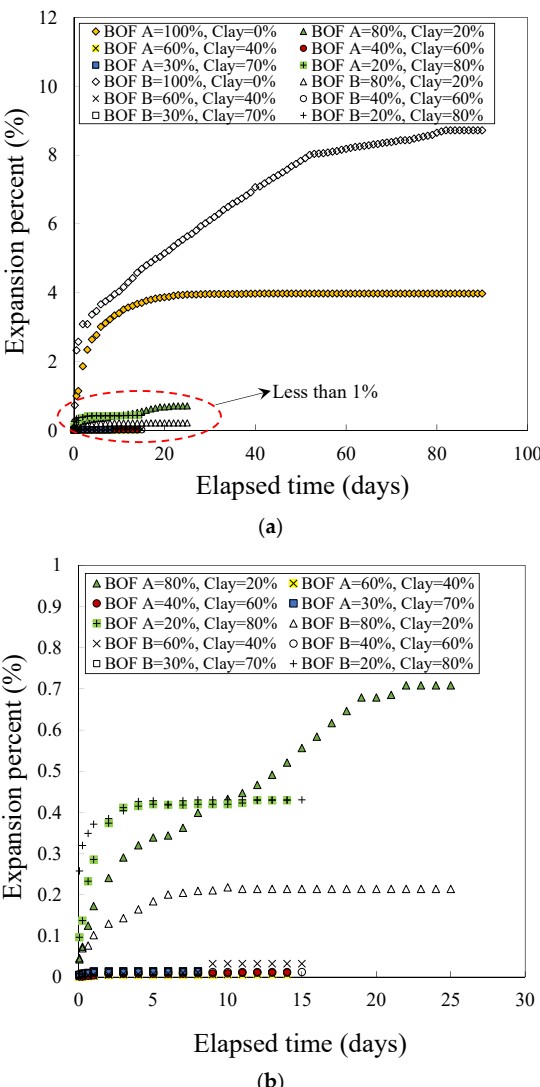

(**a**)

(**b**)

**Figure 5.** Expansion percentage according to the BOF slag/clay ratios of different slag types: (**a**) expansion values for all samples and (**b**) expansion values for samples less than 1%.

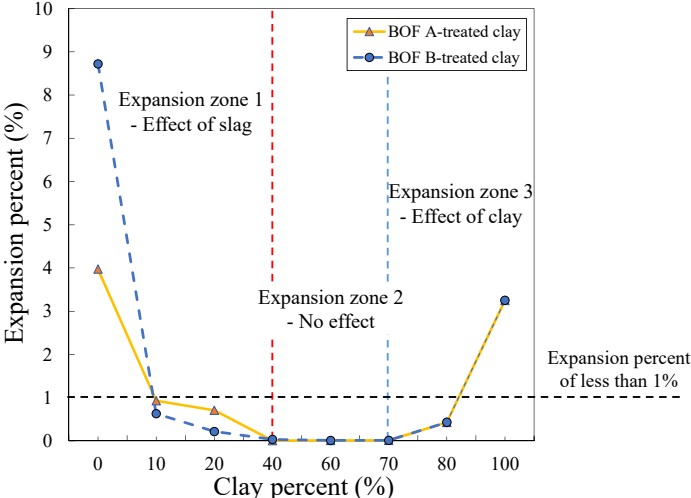

**Figure 6.** Expansion characteristic of BOF-treated clay based on clay percent.

Note that further study is required for various BOF slag and clay contents because expansion behavior could be altered by the type of BOF slag and clay.

### 3.3. Microstructural Properties

Hydration, pozzolanic, and chemical reactions of binders, such as cement and lime, produce novel cementitious compounds (C-S-H and C-A-H gels) within the soil textiles. These novel compounds increase interparticle contact and fill the pore spaces in the soil fabric, thereby hardening the mixture. Kang et al. [2] mentioned that the stabilization mechanisms of the BOF-treated clay are the hydration reaction of the clinker that is contained in the BOF slag and pozzolanic reactions between calcium hydroxide ($Ca(OH)_2$) and Si/Al from soil. In addition, they observed the reticulation structure of the amorphous C-S-H gel and the platy AFm phases with intergrowths of rod-like ettringite within the flat clay structure and flocculated clay–BOF cluster on all samples.

Figure 7 presents microstructural images of the BOF-treated clay with a BOF slag of 40% and setting water content of 1.2 $w_{LL}$, after 28 and 90 days curing, with two additional images taken from [2]. The magnification of the SEM images is about 3000 times (10 ụm) with operation at 20 kV excluding the images (Figure 7e,f) obtained from the literature. In the SEM micrograph of all samples, the cement hydration products, such as the amorphous C-S-H gel, AFm phase, and ettringite, were evenly observed using the clay matrix and flocculated clay–BOF cluster. These cementitious compounds exhibited more significant growth relative to the increase in the curing time (28 to 90 days). This phenomenon triggers an increase in the strength of the BOF-treated clay. In addition, in the BOF B slag case (Figure 7c,d), the reticulation structure of the amorphous C-S-H gel, platy AFm phases, and rod-like ettringite within the flat clay structure and flocculated clay–BOF cluster were significantly denser than those of the BOF A slag (Figure 7a,b).

Figure 7e,f present the microstructural images of the BOF-treated clay acquired from JFE Steel Corporation and Tokuyama port in Japan. In addition, its test conditions included 20% of the BOF slag and a setting water content of 1.5 $w_{LL}$ (water content = 160.7%), after 28 and 90 days curing [2]. The reticulation structure of the amorphous C-S-H gel and platy AFm phases was more significant than that of the samples in this study, although the samples studied in the literature have higher water content and lower BOF slag content than the samples in this study. In addition, the strength magnitude of the sample was also similar to that of the sample in this study. This indicates that there are various factors necessary to develop the strength of the BOF-treated dredged clay, and these factors corresponded to the free CaO content, particle size, and mineralogical composition of the BOF slag, as well as the amorphous Si and Al, water, and organic contents in the dredged clay. Therefore, further study requires the quantitative effect of developing the strength of each factor using different types of BOF slag and dredged clay, and under different conditions.

### 3.4. Environmental Impact
#### 3.4.1. pH Value

To investigate environmental impact, corrosivity measurements obtained by pH values and the concentration quantification of heavy metals were conducted for various samples comprising different types and contents of BOF slag, as well as setting water content. The pH of the BOF-treated clay samples was measured from the leachate water collected for each sample at different curing periods (7, 14, 21, 28, and 90 days) and setting water contents of 1.2, 1.5, and 2.0 $w_{LL}$, and the values obtained are presented in Table 5. Regardless of the curing period and setting water content, the pH values of the leachate water from the raw BOF slags were maintained as high alkaline, ranging from 12.42 to 12.77. Furthermore, the pH values of the BOF-A-treated clay samples at 7, 14, 21, 28, and 90 days showed various ranges of between 10.63 and 12.17, 11.49 and 12.30, 10.82 and 12.31, 10.16 and 10.94, and 9.22 and 11.41, respectively. In BOF-B-treated clay case, the ranges of pH values at the same corresponding curing times were between 11.17 and 11.67, 11.62 and 12.10, 11.35 and 11.89,

10.17 and 11.20, and 9.04 and 11.34, respectively. The pH values of the leachate from treated samples of both BOF slags became smaller than those of raw BOF slags. Interestingly, the pH values of all BOF slag-treated samples were almost kept constant at 21 days curing after placing them in water, then they slightly decreased during the remaining curing periods of 28 and 90 days. Consequently, all BOF slag-treated samples exhibited alkaline features owing to the hydroxide released from the BOF slag; however, their values were satisfied for the limitation (pH range of 2.5 to 12.5) of corrosivity for alkaline samples [37,38]. Therefore, the BOF slag-treated clay exhibits sufficient resistance to corrosivity based on the pH value.

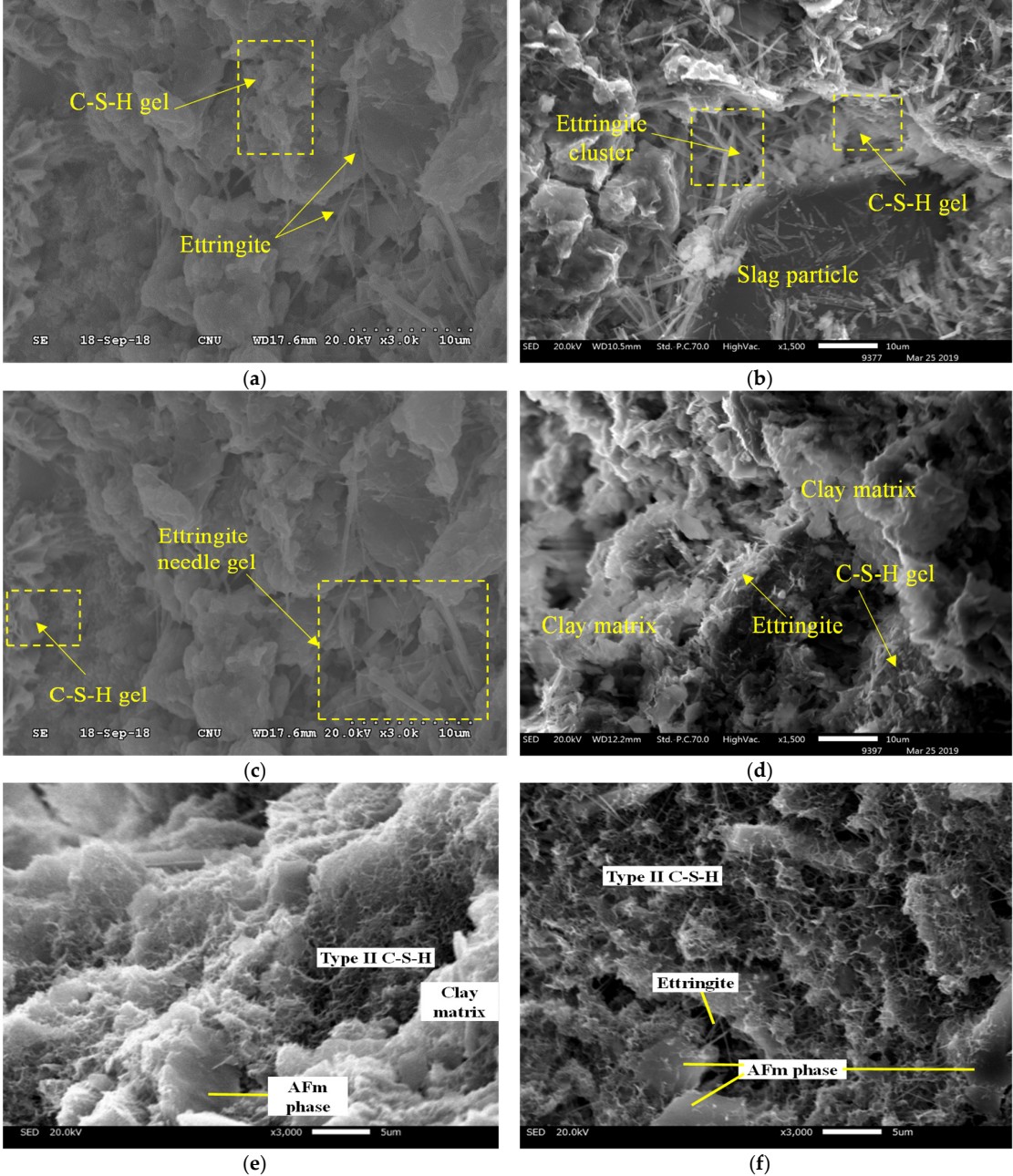

**Figure 7.** SEM micrographs of BOF-treated clays at BOF slag of 40% and water content of 1.2 $w_{LL}$; (**a**) BOF A slag at 28 days curing, (**b**) BOF A slag at 90 days curing, (**c**) BOF B slag at 28 days curing, (**d**) BOF B slag at 90 days curing, (**e**) * 20% BOF slag with water content of 1.5 $w_{LL}$ at 28 days curing, and (**f**) *30% BOF slag with water content of 1.5 $w_{LL}$ at 90 days. * is obtained from Kang et al. [2].

**Table 5.** pH values of BOF slag-treated samples.

| Sample Code | BOF Slag Content (%) | Setting Water Content ($w_{LL}$) | pH Values at Curing Periods (Days) | | | | |
|---|---|---|---|---|---|---|---|
| | | | 7 | 14 | 21 | 28 | 90 |
| | 100 | - | 12.42 | 12.67 | 12.72 | 12.77 | 12.75 |
| | 40 | | 11.15 | 11.92 | 11.37 | 10.94 | 9.22 |
| | 30 | 1.2 | 11.36 | 12.08 | 12 | 10.21 | 9.6 |
| | 20 | | 10.63 | 11.49 | 11.74 | 10.82 | 11.2 |
| | 40 | | 11.62 | 11.94 | 10.82 | 10.9 | 9.46 |
| BOF A | 30 | 1.5 | 11.24 | 11.92 | 11.87 | 10.68 | 10.18 |
| | 20 | | 11.79 | 11.82 | 11.69 | 10.76 | 10.96 |
| | 40 | | 12.17 | 12.3 | 12.31 | 10.16 | 10.35 |
| | 30 | 2.0 | 11.32 | 11.99 | 11.99 | 10.67 | 11.14 |
| | 20 | | 11.19 | 11.77 | 11.8 | 10.69 | 11.41 |
| | 100 | - | 12.56 | 12.6 | 12.68 | 12.69 | 12.67 |
| | 40 | | 11.47 | 11.62 | 11.62 | 10.86 | 9.21 |
| | 30 | 1.2 | 11.24 | 11.73 | 11.65 | 10.75 | 11.11 |
| | 20 | | 11.3 | 11.71 | 11.71 | 10.97 | 11.17 |
| | 40 | | 11.17 | 11.24 | 11.35 | 10.89 | 9.04 |
| BOF B | 30 | 1.5 | 11.53 | 11.89 | 11.8 | 10.61 | 11.17 |
| | 20 | | 11.49 | 11.71 | 11.59 | 10.33 | 10.99 |
| | 40 | | 11.67 | 12.1 | 11.89 | 10.17 | 10.09 |
| | 30 | 2.0 | 11.61 | 11.96 | 11.65 | 10.44 | 10.57 |
| | 20 | | 11.45 | 11.92 | 11.72 | 11.2 | 11.34 |

### 3.4.2. Leachate Concentration

Furthermore, the concentrations of heavy metals in the BOF-treated clay are very significant in assessing the environmental impact in water as the BOF-treated clay was directly mixed and used with water (seawater and/or groundwater) in the field. Figure 8 presents the leachate concentration of heavy metals for the raw BOF slag and BOF slag-treated samples. The heavy metal contaminants of all samples were compared with the threshold effect levels in the marine sediment environment standard of the Ministry of Oceans and Fisheries in the Republic of Korea, as listed in Table 6. Accordingly, the concentrations of all elements were included within the threshold effects level [39]. Hence, the BOF slag-treated clay can be classified as a nonhazardous material, which has no critical adverse environmental impact; therefore, it can be safely applied as a construction material within seawater regions in the coastal construction field.

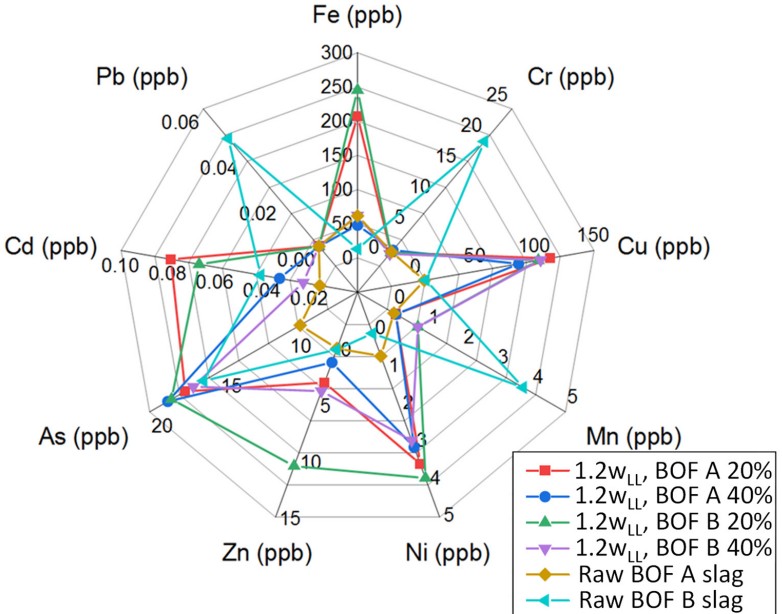

**Figure 8.** Leachable substances in leachate of BOF-treated clay.

**Table 6.** Maximal contamination levels for marine sediment environment.

| Element | Pb | Cr | Cu | Cd | Ni | Zn | As | Mn | Fe |
|---|---|---|---|---|---|---|---|---|---|
| Limits (ppb) | 44,000 | 116,000 | 20,600 | 750 | 47,200 | 68,400 | 14,500 | N.A | N.A |

## 4. Conclusions

In this study, the time-dependent strength behavior, expansion, microstructural properties, and environmental impact of BOF slag-treated clay, which can be adopted as construction and filling material, were investigated with various BOF slag and water contents, as well as at different curing times in South Korea. Based on the results obtained, the following conclusions were drawn:

1.  The strength development of the BOF slag-treated clay can be indicated in three phases depending on the curing time with different incremental strength ratios. These phases include phase I (small-strength increment), phase II (acceleration-strength increment), and phase III (moderation-strength increment).

2.  In phase I, the $D_{50}$ and F-CaO content are responsible for the strength development of the BOF slag-treated clay. In detail, the strength development immediately after mixing at 0.5 h curing was controlled by large particle sizes owing to the increase in effective particle contact area. In addition, the F-CaO content is an important factor influencing the incremental strength ratio owing to the hydration reaction at the early curing time.

3.  In phases II and III, the $FC$ and $D_{50}$ potentially play a dominant role in improving the strength development of the BOF-treated clay. The strength development of the BOF slag-treated clay at long-term curing is attributed to the formation of cementitious hydrates, such as C-S-H and C-A-H gels, which are generated by the reaction between $Ca(OH)_2$ in the BOF slag and the Si and Al content in dredged clays. Here, the smaller particle size evidently has a specific surface area with high reactivity, and thus generates more cementitious hydrates.

4.  The expansion magnitude of the adopted samples is not only influenced by the F-CaO content but also the particle size of the BOF slag. Hence, the contribution to the expansion of the BOF-treated clay can be altered based on the BOF slag and clay contents. Based on the results obtained from this study, the expansion behavior of the BOF-treated clay can be classified into three zones with different expansion percent: expansion zones 1 (effect of slag), 2 (no effect), and 3 (effect of clay).

5.  The pH values of the leachate from the treated samples of both BOF slags became smaller than those of the raw BOF slags. Interestingly, the pH values of all BOF slag-treated samples were almost maintained constant at 21 days of curing after placing them in water. Then, they slightly decreased during the remaining curing periods at 28 and 90 days. Therefore, the BOF slag-treated clay exhibits a sufficient resistance to corrosivity based on the pH value. In addition, the heavy metal contaminants of all samples were compared with the threshold effects level of the marine sediment environment standard of the Ministry of Oceans and Fisheries in the Republic of Korea. Accordingly, the concentrations of all elements were included within the threshold effects level. Hence, the BOF slag-treated clay can be classified as a non-hazardous material, which has no critical adverse environmental impact; therefore, it can be safely used as a construction material within seawater regions in the coastal construction field.

**Author Contributions:** G.-oK.: Conceptualization, Methodology, Investigation, Writing—original draft, Writing—review and editing. J.-g.K.: Conceptualization, Methodology, Investigation, Writing—original draft, Writing—review and editing. J.-y.K.: Investigation, review and editing. Y.-s.K.: Investigation, Writing. All authors have read and agreed to the published version of the manuscript.

**Funding:** This work was supported by a National Research Foundation of Korea (NRF-2018R1C1B600 8095) grant funded by the Korean government (MSIT), and this study was conducted with research funds from Gwangju University in 2021.

**Institutional Review Board Statement:** Not applicable.

**Informed Consent Statement:** Not applicable.

**Data Availability Statement:** The data presented in this study are available on request from the corresponding author.

**Conflicts of Interest:** The authors declare no conflict of interest.

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
