# Peer review of "Time-Dependent Strength Behavior, Expansion, Microstructural Properties, and Environmental Impact of Basic Oxygen Furnace Slag-Treated Marine-Dredged Clay in South Korea"

_sustainability, doi:10.3390/su13095026_

Round 1
Reviewer 1 Report
In this work, authors have investigated the behaviour of dredged clay of marine origin by analysing some of its resulting properties after undergoing a treatment process in a basic oxygen furnace. The aim of this paper is of interest but authors should be improve some aspect:
Section 1. Introduction: Authors have carried out a bibliographic search of the state of the art and clearly indicate the main objective and frame the importance of the study in the proposed scenario. It would be advisable to increase the number of references to show that a thorough search of the subject has been carried out. A correct number of references would be 35 and more.
Section 2. Experimental program: In order to maintain the journal's author guide, I would recommend authors to rename chapter 2 to Materials and Methods, as this would maintain homogeneity with the rest of the journal's scientific works. It would be advisable for authors to add the name of the standards they have followed in some of the essays. As for the rest of the sections in this chapter, they are correctly assigned and described.
Section 3. Results and discussion: If we focus on the environmental impact carried out, it would be advisable from the outset to indicate what type of tests or trials have been carried out specifically, in this case of pH and leachates, given that it could be confused with a more complete study involving an LCA, and this would not be the case.
As an additional note, Figure 5 presents a wide variety of series that make it difficult to read the results correctly even in the enlarged version. Could the way of presenting the data be changed to make it clearer? Through other, more differentiated pictograms, for example?
In Table 5, to avoid repeating the word "days" so many times, it could be simplified with a single "(days)" in the cell: "pH values at curing periods".
Section 4. Conclusion: The conclusions are concise and consistent with the rest of the work presented.
I would like to congratulate the authors for their hard work.
Reviewer 2 Report
In my opinion, it seems to me a good work that can be published. I only find minor corrections or clarifications that need to be made.
I suggest the substitution of the verb “to realize” by others like “to obtain”, “to achieve”, “to accomplish” across the full paper.
Page 4, Line 11. Perhaps this sentence should say something like “The initial water content was calculated as a ratio to the liquid limit”. As it is expressed now it appears as if it refers to an action that is repeated a few times in time, not to a proportion
Page 5, Lines 29 and 35. Perhaps you should use general forms without the definite article “the”. “Samples were prepared…” “Cylindrical specimens…”
Page 6, Line 5. This reference is not correct
Page 7, Line 7. I think that the authors have got the numbers wrong. “…the starting points of strength development at the inactive and active zones corresponding to phases Ⅰ and Ⅱ in this study are 0.5 h to 5 h and 5 h to 7 days, respectively”, from Fig. 2 it would appear that phase I extends from 0.5 h to 10 h and phase II from 10 h to 3 days
Page 8, Lines 2-3. I think that instead “and the initial water content decreased under the same BOF slag content condition.” you wanted to say “and the strength decreased with an increasing initial water content under the same BOF slag content condition”
Page 8, Lines 8-10. The authors try to say that, in the long term, BOF A exhibits less resistance than BOF B, but in my opinion, the idea is not clear. I think that you mean that the BOF A slag samples show a strength approximately 0.25 times LOWER than BOF B samples at later curing times
Page 8, Lines 20-21. Subscript of b are not correct. For Phase II it should say β2 and for Phase III β3
Page 14, Line 11. Avoid passive voice. “the pH values…varied in the range…”
Fig. 8. It would be interesting, if it does not add too much complexity or compromises the legibility of the graph, to include the thresholds according to the mentioned environment standards. Or at least to show them in a Table
Reviewer 3 Report
Comments to the Authors:
The authors of this paper present an interesting investigation of the time-dependent strength behavior, expansion, micro-structural properties, and environmental impact of a mixture of basic oxygen furnace slag and marine-dredged clay generated in South Korea. However, some details should be considered by the authors:
GENERAL COMMENT: Page 1: Although the abstract is well written, it is a bit long. Thus it could be reduced by the authors.
COMMENT: Page 2, line 01: More recent references should be added.
COMMENT: Page 4, line 01: Few details about the X-ray fluorescence experimental setup may be added.
COMMENT: Page 6, line 39 & Fig.2: Comments may also be added concerning the errors introduced in strength behavior measurements. These errors could also be depicted in Fig. 2 (as error bars).
COMMENT: Page 13, Fig.7: The SEM micrographs shown in Figure 7 are not so persuasive. The scale (magnification) of each image is hardly seen. Also, the scale (magnification) of each image is different. I suggest the authors to add images of the same scale, for a direct comparison. Finally, few details about the SEM experimental setup may be added.
The reported data and the results support the authors conclusions. Therefore, I think that this paper can be published (after revision).
